# Spontaneous Lethal Outbreak of Influenza A Virus Infection in Vaccinated Sows on Two Farms Suggesting the Occurrence of Vaccine-Associated Enhanced Respiratory Disease with Eosinophilic Lung Pathology

**DOI:** 10.3390/v16060955

**Published:** 2024-06-13

**Authors:** Wencke Reineking, Isabel Hennig-Pauka, Ludger Schröder, Ulf Höner, Elena Schreiber, Lukas Geiping, Simon Lassnig, Marta C. Bonilla, Marion Hewicker-Trautwein, Nicole de Buhr

**Affiliations:** 1Department of Pathology, University of Veterinary Medicine Hannover, 30559 Hannover, Germanymarion.hewicker-trautwein.ir@tiho-hannover.de (M.H.-T.); 2Field Station for Epidemiology, University of Veterinary Medicine Hannover, 49456 Bakum, Germany; isabel.hennig-pauka@tiho-hannover.de (I.H.-P.); elena.schreiber@tiho-hannover.de (E.S.);; 3Tierärztliche Praxis Peheim, 49696 Peheim, Germany; 4Tierärztliche Praxis in Schöppingen, 48624 Schöppingen, Germany; 5Institute of Biochemistry, University of Veterinary Medicine Hannover, 30559 Hannover, Germany; simon.lassnig@tiho-hannover.de (S.L.); marta.bonilla@tiho-hannover.de (M.C.B.); 6Research Center for Emerging Infections and Zoonoses (RIZ), University of Veterinary Medicine Hannover, 30559 Hannover, Germany

**Keywords:** pigs, swine influenza virus, VAERD, eosinophils, eosinophilic lung pathology, NETs

## Abstract

Influenza A virus (IAV) infections in swine are usually subclinical, but they can reach high morbidity rates. The mortality rate is normally low. In this study, six vaccinated, spontaneously deceased sows revealed IAV infection and enhanced neutrophilic bronchopneumonia with unexpectedly large numbers of infiltrating eosinophils. The purpose of this study was to characterize these lung lesions with special emphasis on the phenotypes of inflammatory cells, the presence of eosinophilic peroxidase (EPO), and neutrophil extracellular traps (NETs). The number of Sirius red-stained eosinophils was significantly higher in the lungs of IAV-infected sows compared to healthy pigs, indicating a migration of eosinophils from blood vessels into the lung tissue stimulated by IAV infection. The detection of intra- and extracellular EPO in the lungs suggests its contribution to pulmonary damage. The presence of CD3^+^ T lymphocytes, CD20^+^ B lymphocytes, and Iba-1^+^ macrophages indicates the involvement of cell-mediated immune responses in disease progression. Furthermore, high numbers of myeloperoxidase-positive cells were detected. However, DNA-histone-1 complexes were reduced in IAV-infected sows, leading to the hypothesis that NETs are not formed in the IAV-infected sows. In conclusion, our findings in the lungs of IAV-infected vaccinated sows suggest the presence of so far unreported field cases of vaccine-associated enhanced respiratory disease.

## 1. Introduction

Influenza A virus (IAV) infections are endemic in herds of domestic swine worldwide [1,2,3,4]. Genetic reassortment of human and/or avian, and/or swine IAVs occurs very frequently and is responsible for the epidemiological situation of influenza in pigs [4].

According to phylogenetic and epidemiological investigations, four main swine IAV subtypes circulate in European swine populations [5,6,7]. The avian-derived virus referred to as H1avN1av was introduced from waterfowl to pigs [2]. Swine H3N2 resulted from the reassortment of porcine H1avN1 with a human seasonal H3N2 [7]. The human-like H1N2 (H1huN2) developed from the reassortment of porcine H3N2 with human-seasonal H1N1. In 2009, the human pandemic IAV (H1N1/2009 H1N1pdm), which was of porcine origin, emerged [5,6]. This virus became endemic and, since 2009, several reassortants including H1pdm subtypes have been detected among pigs in Europe [2,6].

Swine influenza viruses (SIVs) are a major cause of acute respiratory disease of high morbidity in piglets, fattening pigs, and sows [1,4] and, in spite of vaccination, infections with SIVs occur worldwide [2,3]. Clinical signs in typical outbreaks of influenza in pigs include fever, lethargy, dyspnea, nasal discharge, cough, and inappetence [4]. Outbreaks of IAV infections of swine in the field usually are subclinical or mild and can reach a high morbidity rate, but are in general, except from rare reports, of less than 1% mortality [4,8,9,10].

The virulence and pathogenicity of IAV in pigs, however, can be quite variable [4,8,9]. Several reports describe that the inoculation of non-vaccinated pigs with different influenza viruses of porcine, avian, or human origin resulted in non-fatal respiratory disease and lung lesions of varying degrees from only mild to moderate or even severe [11,12,13,14].

Although the severity and extent of lung lesions occurring in pigs after spontaneous or experimental infections with different IAV strains and reassortants can vary considerably, their histopathological features are essentially the same [9,15]. Pneumonic lesions in IAV-infected pigs are mainly characterized by necrotizing bronchitis and bronchiolitis with the presence of desquamated epithelial cells, neutrophils, and macrophages in the lumina of airways and alveoli, and accumulations of lymphocytes around bronchi, bronchioles, and adjacent blood vessels [4,9]. In formalin-fixed paraffin-embedded lung tissue from IAV-infected pigs, viral antigen or RNA can be detected immunohistochemically with mono- and polyclonal antibodies against type A nucleoprotein or in situ hybridization, respectively [16,17].

Vaccination with whole inactivated virus (WIV) with adjuvant is an important tool to control IAV infections in pigs. Application of WIV vaccines in pigs induces the production of neutralizing antibodies to viral hemagglutinin (HA) [18]. The efficacy of such vaccines depends on the homology between the HA of the IAV vaccine and the infection strain, and it is known that vaccine efficacy is reduced against heterologous IAV strains [18,19].

Experiments in pigs have shown that after vaccination with an oil-in-water emulsion adjuvanted WIV and subsequent challenges with heterologous, antigenically mismatched swine IAV strains of the same IAV subtype, vaccine-associated enhanced respiratory disease (VAERD) can occur [20,21]. The pathogenic mechanisms underlying experimentally induced VAERD in pigs are not completely understood. It is discussed that antibody-dependent cellular cytotoxicity (ADCC) involving cross-reactive non-neutralizing antibodies with complement activation and T cell responses leading to the production of pro-inflammatory cytokines likely plays a role in neutrophil recruitment to the lungs and the development of pulmonary tissue damage [20,22,23].

Depending on the stimulus, neutrophils can undergo different mechanisms after infiltrating lung tissue during infections. These include the release of neutrophil extracellular traps (NETs) [24]. During NET formation, neutrophils release a DNA backbone with attached granule proteins like myeloperoxidase (MPO), histones, and antimicrobial peptides [25,26]. NET formation was detected in the lungs of pigs during bacterial infections with *Actinobacillus pleuropneumoniae* (APP) [27] and *Streptococcus suis* [28] and during IAV infection [29]. NETs can be beneficial for the host as they bind and disarm pathogens. However, if they are not sufficiently regulated, an overproduction of NETs can cause detrimental effects to the host [30]. In addition to neutrophils, other cells like eosinophils are described to release extracellular traps (ETs), but to a lesser extent [31].

In humans and animal models, the occurrence of VAERD is associated with the accumulation of eosinophils in the lungs, also known as VAERD with eosinophilic lung pathology. It has been reported for vaccines and vaccine candidates such as respiratory syncytial virus (RSV) and severe acute respiratory syndrome coronavirus 2 (SARS-CoV-2) [32].

In this study, we report spontaneous sudden deaths in six vaccinated, IAV-infected sows with necrotizing bronchopneumonia characterized by the infiltration of lung tissue not only with neutrophils but also with high numbers of eosinophils, suggesting the presence of so-far-unreported field cases of IAV-associated VAERD in pigs. The aim of this investigation was to characterize the lung lesions in these six sows with special emphasis on the participation of eosinophils and mononuclear inflammatory cells.

## 2. Materials and Methods

### 2.1. Animals

Six sows from two different sow farms in northwest Germany were sent to the Field Station for Epidemiology of the University of Veterinary Medicine Hannover in Bakum for diagnostic necropsy due to sudden death. Farm A was a 3000-sow farm with a separated gilt acclimatization unit and areas for breeding and gestating sows. Sows were transported to external farrowing sites. Every 4 weeks, 120 gilts at the age of 14 weeks were purchased from a gilt rearing farm, where gilts had been vaccinated twice in weeks 10 and 14 of life with two commercial WIV vaccines against different swine influenza virus A subtypes (vaccine A (Respiporc FLU3, Ceva, Düsseldorf, Germany): H3N2, H1N1, H1N2, and carbomer as adjuvant; vaccine B (Respiporc FLUpan H1N1, Ceva, Düsseldorf, Germany): H1N1pdm09 and carbomer as adjuvant). After the acclimatization period of three months, gilts were vaccinated with vaccine B at the age of six months and four weeks later, after transfer to the breeding area, with vaccine A. In addition, gilts were vaccinated against porcine parvovirus, erysipelas, porcine circovirus 2 (PCV2), and *Mycoplasma* (*M*.) *hyopneumoniae* at the age of six months and again four weeks later against parvovirus and erysipelas and once against the Porcine Respiratory and Reproductive Syndrome Virus (PRRSV) with a modified live-virus vaccine (PRRSV-1 MLV vaccine). Some days after the second vaccination against swine influenza at the age of 210 days, seven out of 120 gilts died suddenly without showing any clinical signs. Three of these sows were submitted for necropsy to the Field Station for Epidemiology. Within the whole group, individual animals showed slight coughing.

Farm B was a closed farm with 490 sows farrowing in 5-week intervals. Sows were routinely vaccinated against PRRSV using a PRRSV-1 MLV every three months. Three weeks before farrowing, all sows were vaccinated against pandemic influenza virus using vaccine B (H1N1pdm09 and carbomer as adjuvant) as well as an autogenous killed vaccine containing *Escherichia coli*, *Clostridium perfringens* type A, and Rotavirus A. Within a period of four weeks, several sows had shown fever, and eight sows died. Three pregnant sows died one week after PRRSV-1 MLV and influenza virus vaccination and were submitted for necropsy to the Field Station for Epidemiology. The last influenza virus vaccination had been six months previous.

### 2.2. Post-Mortem Examination

Necropsy was performed according to routine diagnostic procedures in the laboratory accredited by the national accreditation body of the Federal Republic of Germany (DIN EN ISO 17025.2017). After assessment of the pig’s carcass, the head was separated from the body for later opening of the cranial cavity and sampling of tonsillar tissue, and the joints were opened and examined. After opening the pleural cavity, the lungs were removed, assessed, and sampled. The right main bronchus was clipped, so that one lung could be rinsed with 400 mL phosphate-buffered saline to gain bronchoalveolar lavage fluid for further examinations. Lung tissue samples including a main bronchus and another one without a main bronchus were harvested from the non-rinsed left lung (cranial and main lung lobe) for bacteriological and molecular biological diagnostics (PCR). In addition, neighboring tissue parts were sampled in 10% neutral buffered formalin (NBF) and were sent to the Department of Pathology, University of Veterinary Medicine Hannover, for histological examination. Routinely, altered lung tissue is sampled with high priority, but in the examined animals, no obvious lung alterations were seen. Subsequently, abdominal organs, including the examination of the gut mucosal surface at various locations, were examined after the opening of the abdominal cavity.

### 2.3. Bacteriological Examination of Lung Tissue

Lung tissues, sampled for bacteriological examination, were processed immediately after necropsy. Samples for PCR diagnostic were either frozen until testing or also immediately analyzed.

The routine standard operating procedure for the diagnostic of bacterial pathogens for samples from the respiratory tract is based on knowledge of clinical veterinary microbiology. Tissue samples were plated on four conventional agar plates, including chocolate blood agar containing nicotinamide adenine dinucleotide (NAD, Blood Agar No. 2, Becton, Dickinson and company, Sparks, USA) for the culture of *Pasteurellaceae* and *Alcaligenaceae*, CNA blood agar (Becton, Dickinson and Company, Sparks, NV, USA) containing nalidixic acid and polymyxin E for the selective culture of *Staphylococcus* spp. and *Streptococcus* spp., Columbia agar with 5% sheep blood (Becton, Dickinson and Company, Sparks, NV, USA), and Gassner agar (OXOID, Hampshire, UK). After culturing for 24 h at 37 °C under atmospheric conditions or in an 8% CO_2_ atmosphere (chocolate blood agar), bacterial growth was recorded. A second inspection of agar plates was performed after a further growth period of 24 h. Bacterial colonies of interest were subcultured for characterization by their biochemical characteristics.

### 2.4. Molecular Biological Diagnostic of Viral Pathogens

For a PCR diagnostic, RNA was extracted from lung tissue (2 × 25 mg) using a commercially available RNA extraction kit (RNeasy Blood and Tissue Kit, Qiagen GmbH, Hilden, Germany). A pathogen-specific protocol with respect to the master mix and temperature protocol was used. The master mix and the temperature protocol differed between the different pathogens. The master mix protocols and cycling conditions for PRRSV and swine influenza A virus were conducted as previously described [33,34].

The PCR diagnostic of PRRSV-1 (EU) and -2 (US) as well as swine influenza A virus was based on commercial PCR diagnostic kits. The use of the PRRSV RT-PCR (EZ-PRRSVTM MPX 4.0 assay, Tetracore R, Rockville, MD, USA) allowed the differentiation of PRRSV-1 and -2 [33]. The utilized RT-PCR (EZ-Universal Flu A 2.0 RT-PCR, Tetracore R, Rockville, MD, USA) allowed the detection of all known swine influenza subtypes [34].

Amplification was performed in a real-time cycler (Applied Biosystems 7500 Real Time PCR System, Thermo Fisher Scientific, Waltham, MA, USA).

### 2.5. Histopathology

Two tissue samples of the lungs of the six sows from cranial and main lung lobes were fixed in 10% NBF for at least 24 h. Thereafter, samples were trimmed, dehydrated, and embedded in paraffin wax according to routine protocols. Sections of 2–3 µm thickness were cut and stained with hematoxylin–eosin (H&E) as well as Sirius red according to routine protocols [35] and subsequently evaluated by two certified, experienced pathologists (MHT, WR).

In addition, paraffin blocks with NBF-fixed lung tissue from pigs originating from eight different farms affected by either IAV (Table 1, farm C-G) or porcine pleuropneumonia (APP, Table 1, farm H-J) in various age groups were selected from the archives of Department of Pathology and the Field Station for Epidemiology (Table 1). In cases of swine influenza, infection with IAV had been proven with the PCR of native lung tissue. For all APP cases, infection with *Actinobacillus pleuropneumoniae* had been detected by bacteriological culture of native tissue.

Furthermore, paraffin blocks with normal lung tissue fixed in NBF were selected from six healthy control pigs of previous animal experiments (Table 1, farm RIZ). These samples were collected from the left *Lobus cranialis*. This animal experiment was approved by the authorities in the committee on animal experiments of the Lower Saxonian State Office for Consumer Protection and Food Safety (Niedersächsisches Landesamt für Verbraucherschutz und Lebensmittelsicherheit (LAVES), Lower Saxony, Germany, registration number 33.8-42502-04-18/2879) [36]. No animal was euthanized for the purpose of this study.

For the semiquantitative evaluation of histopathological lung lesions, H&E stained slides were analyzed according to a previous report about experimentally induced VAERD in pigs [37] (Appendix A
Table A1). Scores were based on the most severe lesion within all lung sections of one animal for necrosis and regeneration of bronchi and bronchioles, peribronchiolar and perivascular cuffing, as well as vascular lesions, such as endothelialitis or perivascular edema, independent from the number and/or area of affected tissue. The percentage of affected tissue over all sections of pulmonary tissue was evaluated and diagnosed with a 4-tier scheme for bronchitis, edema, and hemorrhage (Appendix A
Table A1).

#### 2.5.1. Quantitative Evaluation of Peribronchial and -Bronchiolar Eosinophils in Normal and Inflamed Lungs

Sirius red-stained lung sections were scanned with an Olympus VS200 (Olympus Deutschland GmbH, Hamburg, Germany) with transmitted light at 20× magnification and subsequently analyzed with QuPathsoftware version 0.4.3 (evaluated area) and ImageJ version 1.5.4 (number of eosinophils). Numbers of eosinophils per counted area were normalized to the number of eosinophils per mm^2^ (EGC/mm^2^). Data sets were compared statistically with normal porcine lung tissue as well as the porcine pleuropneumonia cases with the GraphPad Prism version 10.0.2 (232) (GraphPad Software, San Diego, CA, USA).

#### 2.5.2. Immunohistochemistry

Porcine lung samples were deparaffinized and rehydrated in a decreasing alcohol series. Thereafter, either enzymatic or heat-induced antigen retrieval was performed. Slides were blocked with normal goat serum before the application of primary antibodies. Primary antibodies (eosinophilic peroxidase (EPO), influenza A nucleoprotein (IAV NP), CD3, CD20, Iba-1) were incubated overnight at 4 °C and subsequently labeled with appropriate biotinylated goat anti-mouse or goat anti-rabbit preabsorbed secondary antibodies (Abcam, Berlin, Germany) for 45 min at room temperature. Similar IgG concentrations of rabbit or mouse sera were used as negative control stainings. Porcine lymph nodes served as positive controls for CD3, CD20, EPO, and Iba-1. An influenza-infected pig lung (animal no. F2) was used as a positive control for IAV NP. Tissue pretreatment and primary antibody dilutions are listed in Table 2. Immunolabelling was visualized with the VectorStain peroxidase kit and 3, 3′ diaminobenzidine as chromogen according to routine protocols [38].

#### 2.5.3. Evaluation of Extracellular Traps (ETs) in Affected Lungs of Sows

The paraffin-embedded lung tissue sections (2–3 µm thick sections) were analyzed in batches of 10 lung slices at the same time. Sections were stained for immunofluorescence analysis as described previously [39]. Briefly, ETs were stained using the following as primary antibodies: a mouse anti-DNA/histone 1 (IgG2a; MAB3864; Sigma Aldrich, Millipore 1.5 mg/mL diluted 1:273, Billerica, MA, USA) and a rabbit anti-human myeloperoxidase (IgG; A039829-2 Agilent, Santa Clara, CA, USA, 4 mg/mL, 1:364). A respective isotype control was included in each staining batch: murine IgG2a (from murine myeloma, M5409-1 mg conc. 0.2 mg/mL, 1:36.4 Sigma Aldrich, Munich, Germany) and rabbit IgG (from rabbit serum, Sigma Aldrich, Munich, Germany, I5006, 1.16 mg/mL, 1:105.5). All antibodies were incubated overnight at 4 °C.

As secondary antibodies, goat anti-mouse Alexa 488Plus (# A32723, 2 mg/mL, Invitrogen, Carlsbad, CA, USA) and goat anti-rabbit Alexa 568 (# A11011, 2 mg/mL, Invitrogen, Carlsbad, CA, USA, 2 mg, Waltham, MA, USA), both diluted 1:500 in blocking buffer, were used. All samples were processed using the TrueVIEW autofluorescence quenching kit (# SP-8400-15, Vector Laboratories, Newark, CA, USA) following the manufacturer’s instructions and counterstained using Hoechst 33342 (1:1000, stock 50 mg/mL, Sigma Aldrich, Munich, Germany).

The samples were imaged using a Leica TCS SP5 AOBS confocal inverted-base fluorescence microscope with an HCX PL APO ×40 0.75–1.25 oil immersion objective. The settings for each batch were adjusted to their respective isotype controls. Per lung slice, 10 images were randomly taken without overlap. The raw integrated density was measured using ImageJ software (ver. 1.51) for the three different color channels (blue = DNA [counterstaining], green = DNA/histone-1-complexes, and red = MPO). The raw integrated density of both antibody signals was then set in relation to the raw integrated density of the counterstaining.

Data were analyzed using Excel 2016 (Microsoft) and GraphPad Prism 10.0.2 (GraphPad Software, San Diego, CA, USA). Data are presented with mean ± SD as described in the figure legends. The differences between groups were analyzed as described in the figure legends (* *p* < 0.05, ** *p* < 0.01, *** *p* < 0.001).

## 3. Results

### 3.1. Gross and Histopathological Findings

At necropsy, the lungs of the six sows showed no retraction and alveolar edema. Due to extended interlobar septae, interstitial pneumonia was suspected.

The three IAV-positive pigs from the archive of the Department of Pathology exhibited poorly retracted and hyperemic lungs as well as hyperplastic and edematous pulmonary lymph nodes.

APP lungs showed poor retraction as well as multifocal to coalescing areas of hyperemia, atelectasis, and consolidation. The visceral pleura was multifocally attached to the pleura of the thoracic cavity. Pulmonary lymph nodes were enlarged and edematous.

A histological examination of the lung tissue of all six sows (Appendix A
Table A2) revealed the presence of necrotic airway epithelium with bronchitis and bronchiolitis associated with the infiltration of numerous neutrophilic and eosinophilic granulocytes. In the lungs of all six sows, there was mild broncho-interstitial mononuclear pneumonia. Furthermore, peribronchial lymphocytic cuffing was found in all six sows, and the majority of animals showed mild to severe perivascular infiltrates of mononuclear cells (Figure 1). Semiquantitative scoring revealed that all lesions varied from mild to severe between the lung tissue samples of the individual animals (Appendix A
Table A2).

The six IAV-infected pigs of younger age groups (Table 1, farm C-G) showed comparable but less severe changes in lung parenchyma. Vascular changes were especially less severe in these animals.

The six APP pigs (Table 1, farm H-J) with pleuropneumonia showed different stages of the disease. In some areas, hyperemia and proteinaceous fluid admixed with some neutrophils were the most prominent features, while in other areas, fibrinous exudate and neutrophils were accentuated. In addition, areas with complete loss of normal tissue architecture due to coagulative necrosis were demarcated by moderate to high numbers of viable and degenerated neutrophils. Throughout the sections, varying numbers of oat cells were present.

In the lung tissue of six healthy control pigs, except from multifocal to diffuse atelectasis and/or emphysema, no histological abnormalities were found.

### 3.2. Quantification of Neutrophil Extracellular Traps (NETs) in Lung Tissue Sections

Using confocal immunofluorescence microscopy, NET-formation was analyzed in lung tissue sections with quantification of DNA-histone-1 complexes and MPO. The immunofluorescence intensity for both markers was calculated in relation to the staining of DNA (blue), as this reflects the number of nuclei. The values for DNA-histone-1-complexes were significantly reduced in IAV-infected pigs of younger age groups and IAV-infected sows, whereas a significantly increased intensity of the MPO signal was detected in the IAV-infected sows (Figure 2).

### 3.3. Quantification of Eosinophilic Granulocytes in Sirius Red-Stained Lung Tissue Sections

Infiltration of eosinophils in all diseased animal groups was accentuated around the conductive airways and vasculature (Figure 3), while low numbers of eosinophils were distributed randomly throughout the parenchyma in all animals of the unaffected control group. In all pigs of the diseased groups, varying numbers of eosinophils were detected in the alveolar septa, with the highest numbers in the IAV-infected sows. In one sow (animal no. B3), Sirius red staining was not evaluable because of repeatedly reduced staining quality and intensity.

Quantification and statistical evaluation of the number of eosinophils in the lung tissue of the four groups of examined pigs revealed that their number was significantly higher in the lungs of the six IAV-infected sows than in pigs with pleuropneumonia (*p* = 0.02) and in healthy control pigs (*p* = 0.0068) (Figure 4, Appendix A
Table A3). The tissue sections of six sows yielded a mean count of 48.74 cells/mm^2^ (range: 12.2–126.5), while in the six IAV-infected pigs of younger age groups, a mean count of 12.28 cells/mm^2^ (range: 2.7–30.3) was detected. The lowest mean values were detected in APP pigs with 7.2 cells/mm^2^ (range: 0.15–26.3) and the healthy controls with 4.78 cells/mm^2^ (range: 1.8–12.2).

### 3.4. Results of Immunohistochemistry for EPO

EPO distribution resembled the staining pattern of Sirius red-positive cells. In addition, the deposition of EPO was detected in the perivascular and/or peribronchiolar areas of all four groups (Appendix A
Figure A1).

### 3.5. Phenotyping of Pulmonary Inflammatory Cell Infiltrate

All animals of IAV, APP, and healthy control groups showed mild to moderate infiltration of CD3^+^ T lymphocytes and lower numbers of CD20^+^ B lymphocytes. In the six sows, a slight increase in T and B lymphocytes was seen (Appendix A
Figure A2A,B).

Moderate to severe numbers of Iba-1^+^ macrophages were detected in all diseased animals, but slightly less in the healthy controls (Appendix A
Figure A2C, Appendix A
Table A4 and Table A5).

### 3.6. Immunohistochemical Detection of IAV NP Antigen in Lung Tissue

IAV nucleoprotein antigen was only detected in three ‘IAV pigs’ (F1, F2, G1) and was mainly found within the nuclei of low numbers of bronchial and bronchiolar epithelial cells, especially within affected bronchi and bronchioles. Occasionally, positive nuclei were also present in the peribronchial cuffs. No animal of the ‘IAV sows’, APP, and healthy control groups showed IAV positivity (Appendix A
Figure A2D).

### 3.7. Virological and Bacteriological Results

The selection of further diagnostic procedures in animals sent for routine diagnostics was dependent on the anamnestic report and organ alterations detected during necropsy. Bacterial pathogens were not found in five sows. Only from the lung tissue of one sow (pig B2), β-hemolytic *streptococci* were cultured in low amounts. The lung tissue of all sows was tested for PRRSV with negative results.

Due to the fact that PCV2 vaccination was performed on all farms and no typical organ alterations were present (e.g., swollen lymph nodes, wasting), only two gilts from farm A were tested for this pathogen. The subtyping of IAV from BALF or lung tissue was not performed in all sows from the same farm due to cost reasons. The success of subtyping from tissue samples depends on viral load and is not always successful. Diagnostic findings are summarized in Table 3.

## 4. Discussion

In this study, six sows from two different farms in northwest Germany had been submitted for necropsy because of sudden death. Histopathology revealed the presence of extended necrotizing bronchopneumonia. Using PCR and immunohistochemistry on lung tissue samples, IAV RNA was found in all six animals, but no intralesional IAV NP antigen was detected.

The clinical outcome of infections of swine with IAVs is not only affected by the immune status but also different factors such as age, infection pressure, climatic conditions, homing, and secondary infections with bacteria or other viruses [4]. Outbreaks of IAV infections in swine are usually of low mortality [4,8,9]. There are only a few reports about spontaneous outbreaks of fatal IAV infections with severe lung lesions in pigs in the field [10,40]. In the report of Ma et al. [10], 10% mortality in a herd of finishing pigs was due to a highly virulent triple reassortant H1N1 virus and was characterized as moderate, subacute to chronic bronchopneumonia, and from the lungs of dead animals, in addition to the influenza virus, PRRSV and bacteria, including *Pasteurella multocida* and *Streptococcus suis*, were isolated. In the study of Madec et al. (1989), a severe outbreak in sows was described from which the H1N1 virus was isolated and in which necropsy revealed extended lesions of bronchopneumonia and co-infection with *Pasteurella multocida*. In the present investigation, no co-infection of IAV with bacteria and/or additional viruses was detected in 7/8 sows. In one sow, β-hemolytic streptococci were detected in unaltered lung tissue.

Presumably, the sows examined in this study died from a breakthrough infection with IAV, although they had been vaccinated against IAV according to routine vaccination protocols with commercial WIV vaccines, i.e., a trivalent vaccine (H3N2, H1N1, H1N2), on one of the farms and with a monovalent vaccine (H1N1pdm09) on the other farm. The efficacy of WIV vaccines depends on the homology between the HA of the IAV vaccine and the infection strain. It is known that vaccine efficacy is reduced against heterologous IAV strains [18,19]. Furthermore, the application of such inactivated IAV vaccines cannot completely protect pigs against breakthrough infections because of genomic reassortment and the continuous genetic and antigenic diversity of IAV subtypes [18,41].

The histopathological character and the severity of the lung alterations found in the six sows of this investigation resemble findings described in reports about pigs with experimentally induced VAERD [20,21]. The histopathological features of bronchopneumonia occurring in pigs with experimentally induced VAERD are similar to those described for spontaneous porcine IAV infections, but lung lesions are of increased severity [9,20,21].

In contrast to reports about spontaneous IAV infections and experimentally induced VAERD in pigs, the most remarkable finding of this study was that not only neutrophils but also large numbers of Sirius red-stained eosinophils were present in the altered lung tissue of the IAV-infected sows. Sirius red-stained eosinophils were also present in the lungs of the healthy control pigs, but quantification and statistical evaluation clearly showed that their number was significantly higher in the lungs of the six IAV-infected sows.

In addition to blood eosinophils, tissue-resident eosinophils are regularly present in several tissues [42,43]. In humans and mice, they occur under steady-state conditions in the lungs, thymus, intestines, mammary glands, uterus, and adipose tissue [43,44,45].

Detailed information about the existence and number of tissue-resident eosinophils in the tissues of pigs is missing.

In this study, the lowest number of Sirius red-stained eosinophils was present in the histologically unremarkable lung tissue of the group of six healthy control pigs. Although the number of eosinophils varied between their lungs, which could be due to the low number of animals examined, our results suggest that they represent a physiological population of tissue-resident eosinophils as described for humans and mice [44,45].

As cells of the innate immune system, eosinophils are involved in immune homeostasis, immunity, and immune regulation [46], and their number increases in blood and tissues during specific immune responses and allergic diseases [45]. In response to infectious stimuli, including parasites, bacteria, fungi, and viruses, they can leave the bloodstream, traffic into inflammatory sites, and modulate the immune response [42,43].

In the literature, information about porcine eosinophils seems to be limited and mainly restricted to the characterization of the granule proteins of blood eosinophils [47] and studies on the release of their mediators during eosinophilia following the sensitization of pigs with *Ascaris suum* allergens [48]. In this study, there were no signs of parasitic infection as a possible cause for the presence of eosinophils in the lung tissue of the six sows or the lung tissue of the three different groups of control pigs. However, it cannot be excluded that the presence of eosinophils in the lung tissue of the six healthy control pigs may have been, at least in part, due to housing conditions, e.g., induced by contact with air pollutants such as allergens.

Our investigation revealed that the number of Sirius red-stained eosinophils was significantly higher in the lungs of all six cases of IAV-infected sows in comparison to the lungs of healthy control pigs. These results indicate that the majority of eosinophils in the lungs of the six IAV-infected sows represented cells that had migrated from blood vessels into the lung tissue stimulated by the influenza virus infection. Sirius red-stained eosinophils were also detected in the lungs of the control group of APP-infected pigs and the lungs of the control group of IAV-infected pigs. Possibly, the eosinophils found in the lung tissue of the two latter groups of pigs were attracted to the lung tissue due to infection with APP and IAV, respectively, although it cannot be excluded that the population of eosinophils in their lungs included tissue-resident eosinophils as well.

In reports about histopathological characteristics of bronchopneumonia in pigs with spontaneous or experimental IAV infection including VAERD [4,9,20,21], the participation of eosinophils as part of the inflammatory cell population was not described. In those experiments, VAERD was induced after vaccination with an oil-in-water emulsion adjuvanted WIV and subsequent challenge with a heterologous, antigenically mismatched swine IAV strain of the same IAV subtype [20,21]. The efficacy of WIV influenza vaccines depends on the homology of the IAV vaccine and the IAV infection strain and the effective production of neutralizing antibodies by the host [49]. In the last decades, novel IAV strains and cocirculating variants of the same subtype emerged in the European swine population [50]. Phylogenetic analyses revealed various hemagglutinin/neuraminidase combinations and more than 30 distinct swIAV genotypes as consequences of reassortments [6]. The IAV strains detected in the two farms were not genetically characterized for similarity with the used vaccine strains with respect to their hemagglutinin. One vaccine contained an avian-like H1N1 (A/sw/Haselunne/IDT2617/2001(H1N1) and a strain with human-like H1 (Bakum/1832/2000 (H1N2)). The vaccine, which was used on both farms, contained the pandemic strain A/Jena/VI5258/2009(H1N1)pdm09, in which the hemagglutinin was of swine virus origin [51]. Both IAV strains detected in the farms contained a hemagglutinin of avian origin (H1avN1, H1avN2), so a high heterogeneity between the vaccine and field strain can be assumed. It can be hypothesized that VAERD occurred in vaccinated pigs that were challenged at a later point in time by antigenically diverse IAVs. VAERD might have been mediated by nonprotective antibodies of low avidity in these animals.

The possible risk and occurrence of immunopathological reactions termed VAERD associated with vaccine candidates, which may be associated with eosinophilic lung pathology, has been reported in humans and animal models [32,52,53]. In the 1960s, VAERD occurred in children following vaccination with whole formalin-inactivated RSV vaccine and subsequent exposure to respiratory syncytial virus (RSV) [53,54]. In those cases, a severe inflammatory response occurred in the lungs, which was characterized not only by the presence of neutrophilic granulocytes, lymphocytes, and macrophages but also by numerous eosinophilic granulocytes [53,54]. VAERD associated with eosinophilic lung pathology is also a known problem in the research for vaccine development in human medicine, such as for infection with RSV and severe acute respiratory syndrome coronavirus 2 (SARS-CoV-2) in hamsters and mice [52,53,54,55].

Based on the results from those vaccination experiments, two immunopathological mechanisms involved in the development of VAERD have been identified. One of them involves antibodies without neutralizing activity leading to immune complex formation and deposition and the activation of complement and production of inflammatory cytokines, as demonstrated in the 1960s RSV vaccine trials in children [53]. A second pathway involves a biased T helper 2 (Th2)-like immune response resulting in inflammation with the production of a Th2-like cytokine spectrum with increased levels of interleukin-4 (IL-4), IL-5, and IL-13 leading to the development of VAERD with eosinophilic lung pathology [32,53,56].

The influx of eosinophils to the lungs was also described in sporadic reports about humans with spontaneous IAV infections, in which acute eosinophilic pneumonia associated with eosinophilia in peripheral blood or bronchoalveolar lavage fluid was diagnosed [57,58,59]. Furthermore, an influx of eosinophils into the lungs was demonstrated in a mouse model in which vaccinated animals, after challenges with mismatched influenza virus strains of the same subtypes, had influenza breakthrough infections [60,61].

The most remarkable result of this study was the presence of large numbers of eosinophils within the inflamed lung tissue of the six vaccinated sows that suddenly died from an IAV infection. The post-mortem results of the available tissue samples from the IAV-infected sows did not allow any conclusions regarding the possible mechanisms responsible for this unusual finding. However, we cannot exclude that the occurrence of spontaneous IAV-related VAERD associated with a Th2-biased immune response and production of Th2 cytokines was responsible for the attraction of eosinophils to the lungs of these animals.

So far, the occurrence of VAERD in pigs related to IAV was only described under experimental conditions [20,21,23,62], but not in pigs housed for commercial purposes [49]. In pigs with experimentally induced IAV-related VAERD, it is discussed that non-neutralizing antibodies, subsequent immune complex formation with complement activation as well as excessive expression of pro-inflammatory cytokines, i.e., TNF-α, IL-1β, IL-6, IL-8, and chemokines are involved in the development of enhanced pneumonia and neutrophilic infiltration of lungs [20,62,63].

The histopathological examination of the lung tissue of the IAV-infected pigs of this investigation showed that the inflammatory cell population in the lungs, in addition to neutrophils and eosinophils, also included mononuclear cells, i.e., lymphocytes and macrophages. Phenotypical characterization using immunohistochemistry revealed that CD3^+^ T lymphocytes, CD20^+^ B lymphocytes, and Iba-1^+^ macrophages were present in the inflamed lung tissue of the six sows of this study. A similar composition of inflammatory cells was described in VAERD in children vaccinated with the RSV vaccine [54]. These findings suggest that cell-mediated immune responses may contribute to enhanced disease in pigs with IAV-related VAERD, as already discussed by Gauger et al. (2012).

Using immunohistochemistry on lung tissue sections of the six examined sows, the presence of EPO was demonstrated in the cytoplasm of numerous eosinophils and also in extracellular tissue locations.

Eosinophils are known to contain a variety of mediators such as cytokines, chemokines, extracellular mitochondrial DNA traps (EETs), lipid mediators, and cytotoxic cationic granule proteins, e.g., major basic protein (MBP) and EPO [42,43,46,47]. However, in addition to detrimental effects, certain mediators of eosinophils can have beneficial effects, e.g., anti-infectious and anti-inflammatory effects [43,46]. In respiratory diseases induced by RSV and IAV, eosinophils participate in immune responses by interacting with different immune cells [64]. The results obtained in a combined mouse model of allergy and IAV infection suggest that eosinophils following piecemeal degranulation contribute to host cellular immunity by upregulating antigen presentation markers, modulating the function of T lymphocytes and dendritic cells, and promoting the reduction of virus replication [64,65]. Based on the findings of our investigation, it cannot be excluded that, in addition to mediators of neutrophils, the release of EPO and possibly other eosinophil-derived toxic mediators may have contributed to the lung tissue damage and inflammatory changes in the IAV-infected sows.

Research results about VAERD with eosinophilic lung pathology in SARS-CoV-2-vaccinated hamsters and RSV-vaccinated mice have shown that a Th2-biased immune response with secretion of Th2-associated cytokines is required for the influx of eosinophils into the lungs [52,66]. Regarding the pathogenic role of eosinophils, however, studies in RSV-vaccinated mice revealed that certain disease parameters were mediated by a Th2-biased immune response and distinct CD4 T cell subsets, but not by eosinophils alone [32,66]. On the other hand, in mice vaccinated with the influenza virus, in contrast to SARS-CoV-2-vaccinated hamsters, eosinophilic lung pathology was not associated with an overt Th2 cytokine pattern [60,61].

The association of eosinophils with lung lesions in pigs after spontaneous IAV infection in the field or experimentally induced IAV infection including VAERD has not been reported. The findings in the IAV-infected sows of this investigation suggest the occurrence of so far undescribed spontaneous cases of IAV-associated VAERD with eosinophilic lung pathology. Due to the retrospective character of this post-mortem study, the possible pathogenic role of the eosinophils found in the lungs of the IAV-infected sows regarding their contribution to the development of the lung tissue lesions and the possible roles of the eosinophils on the immune response of the infected sows remains unknown.

The examination of NETs showed an unexpected reduction of DNA-histone-1 complexes and a significantly increased amount of myeloperoxidase in the group of IAV-infected sows (Figure 2) compared to the control group. Whereas, in the IAV-infected piglets, the signal of DNA-histone-1 complexes was also significantly reduced, and MPO was not significantly increased. In an earlier study of APP-infected and non-infected pigs, we detected an increased signal of DNA-histone-1 complexes and neutrophil elastase in the lungs of the infected group and an increase in different NET markers in bronchoalveolar lavage fluid (BALF), indicating NET formation during the bacterial infection [27]. NETs can be induced by different stimuli, including pathogens like IAV [30,67]. Recently, we identified different NET markers in the BALF of IAV field-infected pigs and observed a positive correlation between NET markers and IAV viral load. Furthermore, neutrophils releasing vesicular NETs were found in the BALF of IAV-infected pigs [29]. However, there are no studies about NET formation in lung tissue during IAV in pigs, and it is not fully understood how NETs are induced by IAV [68]. The NET induction with IAV seems to be strain dependent [69]. The DNA-histone-1 complex is, together with MPO or neutrophil elastase, a NET marker if they are found colocalized and/or extracellularly. Nevertheless, the DNA-histone-1 complex is also staining cells that are not undergoing NET formation. Therefore, DNA-histone-1 complexes can be detected in healthy control animals. However, in the infected animals, we identified a reduced DNA-histone-1 complex signal, which can be due to tissue damage that is caused by NET components [70,71,72,73]. Therefore, a quantification analysis for NETs in histology sections based on the DNA-histone-1 complex could be a false negative. However, we identified single NET fibers in the histology sections (Figure 2). To understand the influence of NETs during VAERD, further analysis of example cells in BALF would be needed in future studies. In addition, it must be considered that eosinophils can release extracellular traps (ETs) as well [31]. After the first descriptions of eosinophil extracellular traps (EETs) [74], several further studies followed, describing EETs during infections with bacteria [75], fungi [76], and parasites [77]. Furthermore, EETs have been detected in an asthma mouse model with respiratory syncytial virus (RSV) infection [78] and in plastic bronchitis (PB) associated with IAV infection in humans [79]. Therefore, the observed NET fibers could be a product of the identified eosinophils as well.

## 5. Conclusions

In this study, necropsies on six vaccinated sows submitted because of sudden death from two different farms revealed an IAV infection with necrotizing bronchopneumonia characterized by the infiltration of lung tissue, in addition to neutrophils, with large numbers of eosinophils. Our findings suggest the presence of so far unreported field cases of IAV-associated VAERD in swine resulting from a breakthrough infection. Possibly, severe lung disease in vaccinated pigs in the field occurs more often but remains unnoticed in cases in which necropsy and histopathology are not performed. The histopathological findings in the lungs of the sows resemble descriptions of VAERD with eosinophilic lung pathology in humans and animal models associated with vaccine candidates for viruses, i.e., RSV and SARS-CoV-2. Immunohistochemistry revealed the presence of CD3^+^ T lymphocytes, CD20^+^ B lymphocytes, and Iba-1^+^ macrophages, suggesting that cell-mediated immune responses contributed to the enhanced inflammatory lung lesions. Our observations in these field cases of IAV-infected sows show that further research is needed regarding the efficacy of WIV IAV vaccines and the possible immunopathological mechanisms associated with IAV-associated VAERD in pigs, including the influence of ETs released by neutrophils and/or eosinophils.

## Figures and Tables

**Figure 1 viruses-16-00955-f001:**
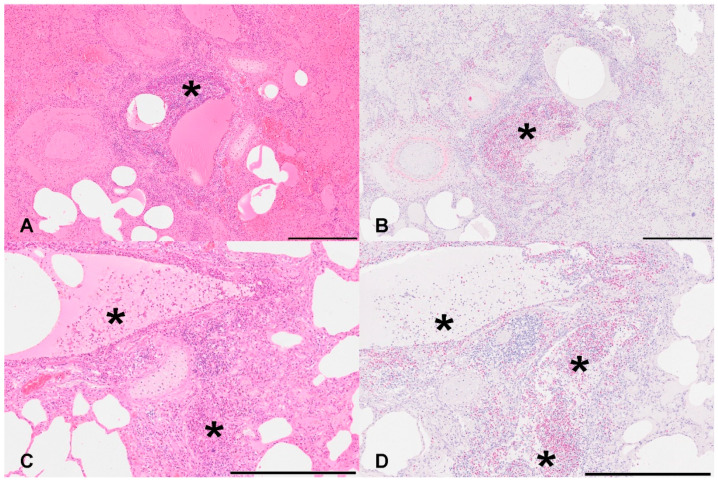
Morphological findings in IAV-infected sows. (**A**,**C**): Accentuated around bronchi and larger vessels, there was a marked infiltration of neutrophilic and eosinophilic granulocytes as well as lower numbers of macrophages and lymphocytes (*). Multifocally, pronounced edema was present. Hematoxylin and eosin. Bar: 500 μm. (**B**,**D**): The presence of eosinophils was confirmed by the Sirius red stainability of numerous granulocytes (*). Sirius red. Bar: 500 μm.

**Figure 2 viruses-16-00955-f002:**
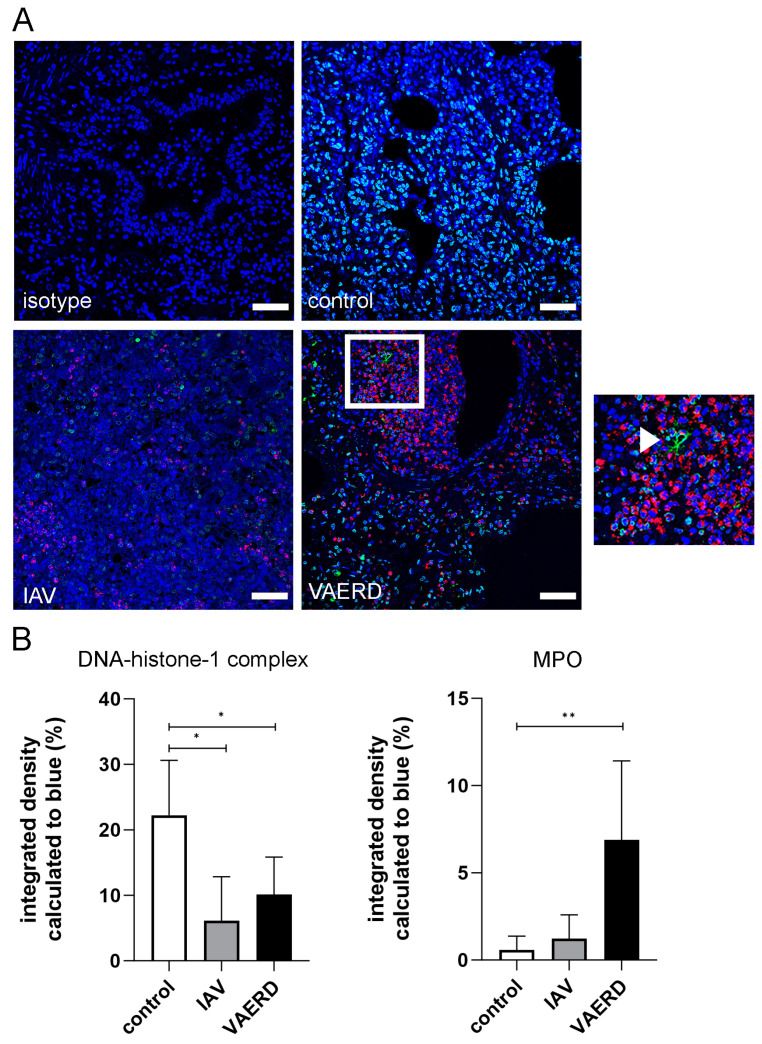
Detection of NETs in lung tissue. (**A**) Representative pictures of immunofluorescence microscopy are shown (blue = DNA (Hoechst), green = DNA-histone-1-complexes, red = MPO). The right image is a magnification from the left image, and the arrowhead marks a NET fiber. The scale bar in all images represents 50 µm. (**B**) Stained lung slices of three to six animals per group were analyzed from two technical runs per animal. Images were analyzed using ImageJ (version 1.5.4) software to determine the percentage of mean fluorescence of the green (DNA-histone-1-complex) and magenta (MPO) signals based on the blue signal (nuclei). The means from the technical runs were calculated and used for statistics. Per animal, 10 immunofluorescence microscopy images were analyzed in each technical run (control n = 6, IAV n = 3, VAERD n = 6 animals). A high intensity of MPO signals was detectable in the lungs of VAERD sows. Data were analyzed with ordinary one-way ANOVA followed by Dunnett’s multiple comparisons test and are presented with mean ± SD (* *p* ≤ 0.05, ** *p* ≤ 0.01).

**Figure 3 viruses-16-00955-f003:**
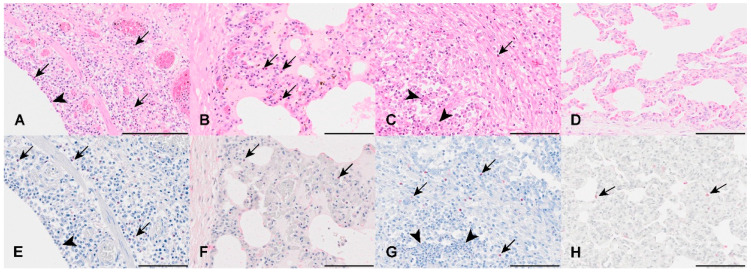
Comparison of eosinophilic count in the different study groups. (**A**,**E**): In IAV-infected sows, numerous eosinophils (arrows) are present within the mucosa and submucosal tissues. Neutrophils are occasionally present within the mucosa (arrowhead). No Sirius red stain was observed in neutrophils ((**E**); arrowhead). (**B**,**F**): The IAV-infected younger pigs have scattered eosinophils (arrows) within the alveolar septa. (**C**,**G**): Low numbers of eosinophils (arrows) are also present within the submucosal tissue and alveolar septa of the APP-infected group. The majority of intraluminal cells were confirmed as neutrophils (arrowheads) by their lack of Sirius red staining (**G**). (**D**,**H**): Single eosinophils (arrows) were present in the alveolar septa of healthy control pigs. ((**A**–**D**): Hematoxylin-eosin; (**E**–**H**): Sirius red stain). Bars: 100 μm.

**Figure 4 viruses-16-00955-f004:**
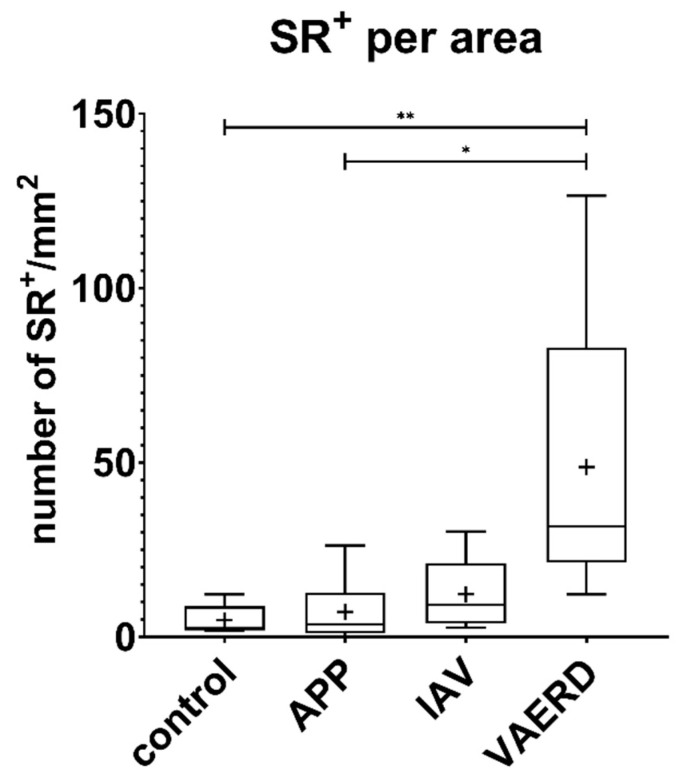
Quantification of Sirius red-positive cells. Increased numbers of Sirius red-positive cells were detected within the IAV-infected sows compared to younger IAV-infected pigs as well as APP-infected pigs and healthy controls. Statistical analysis showed significant differences between the IAV-infected sows compared to healthy controls and APP pigs, but not to the younger IAV-infected group. Min-to-max Box-and-Whisker plot. Kruskal–Wallis test with subsequent Dunn’s test for multiple comparisons. *, *p* ≤ 0.05; **, *p* ≤ 0.01; SR, Sirius red; +, mean count of cells.

**Table 1 viruses-16-00955-t001:** Anamnestic data of pigs included in this study.

Group	Farm	Animal no.	Age Group/ Stage of Production *	Sex	Weight [kg]	Origin
IAV sows	A	1	gilt	Female	121	Bakum
	A	2	gilt	Female	136	Bakum
	A	3	gilt	Female	134	Bakum
	B	1	sow	Female	289	Bakum
	B	2	sow	Female	239	Bakum
	B	3	sow	Female	231	Bakum
IAV pigs	C	1	Nursery pig	Male neutered	6.3	Bakum
	D	1	Suckling piglet	Male neutered	5.6	Bakum
	E	1	Suckling piglet	Male neutered	3.1	Bakum
	F	1	Nursery pig	Male neutered	20.2	Pathology
	F	2	Fattening pig	Male neutered	38.9	Pathology
	G	1	Fattening pig	Male	25	Pathology
APP	H	1	fattening pig	Female	92	Pathology
	H	2	fattening pig	Male neutered	52	Pathology
	H	3	fattening pig	Male neutered	70	Pathology
	H	4	fattening pig	Female	92	Pathology
	I	1	fattening pig	Female	34.6	Pathology
	J	1	fattening pig	Male neutered	32	Pathology
Healthy controls	RIZ	1	Nursery pig	Female	13	Pathology
	RIZ	2	Nursery pig	Female	14	Pathology
	RIZ	3	Nursery pig	Male neutered	14	Pathology
	RIZ	4	Nursery pig	Female	16	Pathology
	RIZ	5	Nursery pig	Male neutered	16	Pathology
	RIZ	6	Nursery pig	Female	15	Pathology

* Suckling piglet: <6 kg and from farrowing pen, nursery pig: ≤25 kg and from nursery unit; fattening pig: 26–100 kg and from fattening unit; gilt: <150 kg, sow >200 kg.

**Table 2 viruses-16-00955-t002:** Primary antibodies used for immunohistochemistry.

Antigen	Host	Clonality	Company	Cat. No.	Pretreatment	Dilution
human CD3	rabbit	polyclonal	Dako	A0452	HIER ^1^	1:200
human CD20	rabbit	polyclonal	Invitrogen	PA5-16701	HIER ^1^	1:300
human eosinophilic peroxidase	rabbit	polyclonal	Abcam	ab238506	HIER ^2^	1:2000
human Iba-1	rabbit	polyclonal	Wako	019-19741	HIER ^1^	1:500
Influenza A nucleoprotein	mouse	monoclonal; clone HB65	Kerafast	FCG013	Proteinase K	1:200

HIER: heat-induced epitope retrieval; ^1^ citrate buffer, pH6, microwave, 20 min; ^2^ Target Unmasking Fluid^®^ (PanPath, Budel, The Netherlands; Cat. SSP-0025-TUF3), steamer, 95 °C, 15 min.

**Table 3 viruses-16-00955-t003:** Diagnostic findings in six sows with VAERD.

Animal Number	IAV detected in Lung Tissue	Additional Findings in Other Organs
A1	+H1avN1	-
A2	+Subtyping not successful	Lymphnode: PCV2 negative
A3	+	Lymphnode: PCV2 negative Meninges: *Streptococcus suis*
B1	+	Pleura/Peritoneum: β-haemolytic streptococci
B2	+	Meninges: *Pasteurella multocida*
B3	+H1avN2	-

## Data Availability

The authors confirm that the data supporting the findings of this study are available within the published article. Raw data were generated at the University of Veterinary Medicine Hannover, Foundation, Hannover, Germany. Derived data supporting the findings of this study are available from the corresponding author, N.d.B, after request.

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
