# Peer review of "Spontaneous Lethal Outbreak of Influenza A Virus Infection in Vaccinated Sows on Two Farms Suggesting the Occurrence of Vaccine-Associated Enhanced Respiratory Disease with Eosinophilic Lung Pathology"

_viruses, 2024, doi:10.3390/v16060955_

Round 1

Reviewer 1 Report

Comments and Suggestions for Authors

This is an interesting manuscript. I agree with the conclusions of this very important finding that suggests the following, as written by the authors: "Possibly, severe lung disease in vaccinated pigs in the field occurs more often but remains unnoted in cases in which necropsy and histopathology are not performed." The study is very well designed. The methods are clearly written in much details. Results and Discussions are elaborated and provide enough data to support the conclusions of the study.  

After a careful read, I have following minor suggestions for the authors.

L24-46: I strongly recommend using the in-text citations. For example, "The avian-derived virus referred to as H1avN1av was introduced from waterfowl to pigs." Please provide the citation at the end of the sentence(s) rather than at the end of the paragraph. 

L54: "low mortality". Any range, in terms of percentage?

L171-172: Was it an internally standardized protocol? Or, was it a previously published protocol? Can it be cited?

L399: T here are

L409: " no other bacteria and no viruses were isolated from the lungs of the animals" The way it is written is a bit confusing for me. I suggest writing it in a way that is more clear to the reader, such as "no other bacteria and viruses, other than IAV, were isolated from the lungs of the deceased animals. 

I hope I got the above statement right. 

Author Response

Answers to the reviewers -viruses-3066351 - first report

Hannover, 10.06.2024

Dear Editor and reviewers

the decision letter for our manuscript was received on June 7, 2024 and was very much appreciated. We would like to thank the reviewers for the constructive comments and suggestions.

We have prepared a revised version of the manuscript, highlighting the changes made using the Tracked Changes function in Microsoft Word, and we have prepared a point-by-point response in order to address the reviewers’ comments.

We would like to thank the reviewers again for their suggestions and think that the manuscript substantially improved by the changes made. We hope that our work is now acceptable for publication in Viruses.

With kind regards,

Nicole de Buhr, on behalf of all authors.

Answers to the specific comments:

  1. L24-46: I strongly recommend using the in-text citations. For example, "The avian-derived virus referred to as H1avN1av was introduced from waterfowl to pigs." Please provide the citation at the end of the sentence(s) rather than at the end of the paragraph.

Answer:

We agree with the reviewer that this should be emphasized and revised the manuscript accordingly.

2. L54: "low mortality". Any range, in terms of percentage?
Answer:

We thank the reviewer for this question. We have clarified that sentence with data from the respective publications and have revised the sentence as follows:

LL 52-55

Outbreaks of IAV infections of swine in the field usually are subclinical or mild and can reach a high morbidity rate, but are in general, except from rare reports, of less than 1% mortality [4,8–10].”

  1. L171-172: Was it an internally standardized protocol? Or, was it a previously published protocol? Can it be cited?

Answer: We thank the reviewer for raising this important question. Indeed, the protocols were already published and we included this information with citations in the methods section:

Lines 172

“The master mix and the temperature protocol differed between the different pathogens. The master mix protocols and cycling conditions for PRRSV and swine influenza A virus was conducted as previously described [33,34].”

  1. L399: T here are

 Answer:

Thanks for highlighting this typo. We have corrected it and proof read the entire manuscript.  

  1. L409: " no other bacteria and no viruses were isolated from the lungs of the animals" The way it is written is a bit confusing for me. I suggest writing it in a way that is more clear to the reader, such as "no other bacteria and viruses, other than IAV, were isolated from the lungs of the deceased animals.

I hope I got the above statement right.

Answer:

Thanks for your suggestions. We clarified the statement as follows:

In the present investigation, no co-infection of IAV with bacteria and/or additional viruses was detected in 7/8 sows. In one sow β-hemolytic streptococci were detected in unaltered lung tissue.”

Reviewer 2 Report

Comments and Suggestions for Authors

The manuscript presents a detailed investigation into the lung lesions observed in vaccinated sows that succumbed to Influenza A Virus (IAV) infection. The study focuses on the phenotypes of inflammatory cells, the presence of eosinophilic peroxidase (EPO), and the exploration of neutrophil extracellular traps (NETs). The research is timely and relevant, considering the economic and animal health implications of IAV in swine herds.

Major Comments:

1. The discussion section provides a thoughtful interpretation of the results, linking them to existing knowledge on IAV infection and immune responses. The hypothesis regarding the lack of NET formation in IAV-infected sows is particularly intriguing and warrants further investigation.

2.There are a few typographical errors throughout the manuscript that should be corrected.

3.It would be helpful to include additional references to support some of the statements made in the discussion section, particularly those related to the role of eosinophils and NETs in IAV infection.

Overall, this manuscript makes a significant contribution to our understanding of the pathological changes associated with IAV infection in vaccinated sows. The findings have important implications for swine health and vaccination strategies. With minor revisions, this manuscript would be suitable for publication in a peer-reviewed journal.

Author Response

Answers to the reviewers -viruses-3066351 - first report

Hannover, 10.06.2024

Dear Editor and reviewers

the decision letter for our manuscript was received on June 7, 2024 and was very much appreciated. We would like to thank the reviewers for the constructive comments and suggestions.

We have prepared a revised version of the manuscript, highlighting the changes made using the Tracked Changes function in Microsoft Word, and we have prepared a point-by-point response in order to address the reviewers’ comments.

We would like to thank the reviewers again for their suggestions and think that the manuscript substantially improved by the changes made. We hope that our work is now acceptable for publication in Viruses.

With kind regards,

Nicole de Buhr, on behalf of all authors.

Answers to the specific comments:

  1. The discussion section provides a thoughtful interpretation of the results, linking them to existing knowledge on IAV infection and immune responses. The hypothesis regarding the lack of NET formation in IAV-infected sows is particularly intriguing and warrants further investigation.

Answer:

Thanks for your appreciation.

  1. There are a few typographical errors throughout the manuscript that should be corrected.

Answer:

Thanks for highlighting. We have carefully revised the spelling and corrected all detected typos.

  1. It would be helpful to include additional references to support some of the statements made in the discussion section, particularly those related to the role of eosinophils and NETs in IAV infection.

Answer:

Thanks for your comment, we have added additional references whenever suitable.

 Lines 547 ff

“In respiratory diseases induced by RSV and IAV, eosinophils participate in immune responses by interacting with different immune cells [64]. Results obtained in a combined mouse model of allergy and IAV infection suggest that eosinophils following piece meal degranulation contribute to host cellular immunity by upregulating antigen presentation markers, modulating the function of T lymphocytes and dendritic cells, and by promoting reduction of virus replication [64,65].”

Lines 570 ff

 “Due to the retrospective character of this post-mortem study, the possible pathogenic role of the eosinophils found in the lungs of the IAV-infected sows regarding their contribution to the development of the lung tissue lesions and the possible role of the eosinophils on the immune response of the infected sows remains unknown.”

 Lines 582 ff

“Recently, we identified different NET markers in BALF of IAV field-infected pigs and observed a positive correlation of NET markers and IAV viral load. Furthermore, neutrophils releasing vesicular NETs were found in BALF of IAV infected pigs[29]. However, there are no studies about NET-formation in lung tissue during IAV in pigs and it is not fully understood how NETs are induced by IAV [68].”

Lines 598 ff

“After the first descriptions of eosinophil extracellular traps (EETs)[74], several further studies followed describing EETs during infections with bacteria [75], fungi [76] and para-sites [77]. Furthermore, EETs have been detected in an asthma mouse model with respiratory syncytial virus (RSV) infection [78] and in plastic bronchitis (PB) associated with IAV infection in humans [79].”